# Mechanical and Self-Healing Performance of Yellow River Alluvial Silt Treated with Composite Flexible Curing Agent

**Zhiyi Sai [1], Lin Wang [2], Hongchao Han [1], Wenjuan Wu [2],\*, Zhaoyun Sun [2], Jincheng Wei [2], Lei Zhang [2], Guiling Hu [3] and Hao Wu [4]**

1. Shandong Hi-Speed Company Limited, Jinan 250014, China; jtgsdqgkj@163.com (Z.S.); 18765863568@139.com (H.H.)
2. Shandong Transportation Institute, Jinan 250102, China; wanglin@sdjtky.cn (L.W.); andersontwo@163.com (Z.S.); weijincheng@sdjtky.cn (J.W.); realchungsdu@163.com (L.Z.)
3. School of Transportation Engineering, Shandong Jianzhu University, Jinan 250101, China; huguilingtech@foxmail.com
4. Qingdao Highway & Bridge Group Co., Ltd., Qingdao 266033, China; z5hao@126.com
* Correspondence: wuwenjuan@sdjtky.cn

**Abstract:** The silt in the Yellow River alluvial plain has low clay content, low cohesion and poor structure. Its stability has always been a difficult problem in the engineering field. In order to improve the engineering properties of the silt in the alluvial plain of the Yellow River, a new type of silt composite flexible curing agent was prepared by using sintered red mud and matrix asphalt as the main materials to comprehensively stabilize the silt. The aim of this study was to investigate the effects of sintered red mud-asphalt composite flexible curing agent on aged mechanical properties of treated silt, in which the replacement levels of the flexible curing agent below 10% by weight are compared. Apart from the compressive strength, the drying shrinkage, low temperature freeze-thaw and high temperature self-healing ability are measured. The test results show that the flexible curing agent has a positive effect on improving the mechanical properties of stabilized silt. The flexible curing agent series exhibit higher compressive strength, better water stability, resistance to freeze-thaw and high temperature self-healing ability, and lower drying shrinkage compared to silt and cement stabilized silt. The preferred dosage 4%~6% of the flexible curing agent is obtained by mechanical property analysis. The SEM images show that the incorporation of the flexible curing agent helps the silt form dense cementation and non-connected microporous structure, that is beneficial to the improvement of water stability and frost resistance. The asphalt component in the flexible curing agent can reorganize and diffuse in the soil, fill the internal pores and micro cracks, and realize the repair of soil damage and structural reinforcement.

**Keywords:** silt; stabilized silt; sintered red mud-asphalt composite flexible curing agent; mechanical properties; microstructure

## 1. Introduction

The silt in the alluvial plain of The Yellow River is mainly distributed in the middle and lower reaches of the Yellow River in Shandong, Anhui, Henan, Hebei, etc. These soils have a high silt content (>60%), lack agglomerate structure, and are highly susceptible to loss [1–3]. Historically, because the Yellow River carried a large amount of sediment from the Loess Plateau, and then experienced the advance and retreat of seawater, the oscillating subsidence of the crust and the repeated swings of the river body, shaping into the Yellow River alluvial and sedimentary plain [4,5]. Due to its special origin, The particle distribution and structure of silt in the Yellow River alluvial plain are unique. Its particle gradation is poor, the powder content is too high, and the clay content is low, which makes it difficult to stabilize the binder and compact. Moreover, the capillary water in the silt can rise high and the capillary effect is strong, which is easier to make moisture

accumulation in roadbed in seasonally frozen areas. It can result in severe frost heave damage and large post-construction settlement [6–9]. These problems make it difficult for silt to be directly used for roadbed filling. At present, Through technical means such as improved solidification, compaction process optimization and engineering protection measures, the large-scale application of silt to roadbed filling can be realized.

Many scholars have carried out a lot of research on the improvement and stability of silt. The traditional solidifying materials of soil are all solid inorganic binders. Good results have been achieved by using lime, cement and fly ash to improve the soil [10]. It can convert loose soil particles into dense cementitious materials through a series of physical and chemical reactions to improve the strength and durability of silt [11–14]. In the long-term engineering practice, people gradually realize that although these traditional solidified materials can improve the strength of silt, their large dry shrinkage and temperature shrinkage are easy to cause crack, resulting in decrease in compressive strength, impermeability, frost resistance and resistance. It is considered to be the material with large shrinkage and poor water stability among various semi-rigid materials [15–17]. In order to make up for these deficiencies, experts and scholars in related fields are actively committed to the research and development of new soil stabilization technologies.

The researches have been made rapid progress in organic compound curing agents, biological enzyme curing agents and composite curing agents. Organic compound-based soil curing agents are generally liquid and are mainly composed of one or more combinations of water glass, epoxy resin, polymer materials and ionic curing agents [18,19]. This kind of curing agent can promote the exchange of charge in soil particles and soil moisture, and then promote the ionic reaction between the two, and finally play a role in weakening the capillary, pores and surface tension water absorption capacity in the soil. It makes the soil easier to drain and consolidate under the action of external force [20–24]. Ding Rui [25] used X-ray diffraction and X-ray photoelectron spectroscopy to prove that the surface of clay particles would undergo chemical reaction with water glass, and speculated that the chemical reaction may increase the cementation between clay particles. He Jun et al. [26] used water glass-alkali slag-slag to solidify silty clay with high water content, and explored the strength characteristics of the solidified silty clay. Zhao et al. [27] utilized different types of ionic curing agents to cure the expansive soil and evaluated the physical properties such as the expansion rate, liquid-plastic limit and linear shrinkage rate of the cured soil. The change of ion concentration in pore water before and after curing of expansive soil was measured by atomic absorption spectrometry. The expansion potential of expansive soil was obtained by the concentration of cations in pore water. The curing effect of different ionic curing agents was determined. Lynn et al. [28] studied the mechanism of ionic curing agent strengthening montmorillonite by means of chromatography, X-ray diffraction, and titration analysis.

The biological enzyme curing agent is liquid and formed by fermentation of organic matter. Catalyzed by biological enzymes, the adhesion between soil particles will be strengthened when the soil is mechanically compacted, thereby improving the soil engineering properties. The commonly used biological enzyme curing agents mainly include Terrazyme and Permazyme. Greeshma et al. [29] treated high liquid limit montmorillonite from Kerala, India with different concentrations of tyranase and conducted unconfined compressive strength tests on the improved soils with different curing times. The best dosage of the biological enzyme was obtained. Cheng et al. [30] explored the effects of urease concentration, ambient temperature, oil pollution and freeze-thaw cycles on the urease-induced calcite precipitation process through experiments. It proved that this precipitation mode significantly improved the unconfined compressive strength of the soil and its durability under freeze-thaw cycle erosion. Sun [31] used Pyase to solidify Shanghai mixed fill and obtained the change law of the compactness and strength of the soil and gave the optimal dosage of Pyase to solidify the soil. In addition, Peng et al. [32] used the enzyme to solidify coarse sand, fine sand, surface sand and sandy loam, and studied the

strength characteristics of the four kinds of enzyme-solidified soils through the unconfined compressive strength test.

Composite soil stabilizers are new types of solidifying materials, generally in solid and liquid forms, and are prepared from two or more compounds in a certain proportion [33–35]. Dong Jinmei et al. [36] used cement-modified polyvinyl alcohol (SH) to solidify light soil and discussed the influence of age, SH content and soil composition on the strength characteristics of solidified light soil. Liu Chengbin [37] used slag composite curing agent to solidify saline soil and evaluated the unconfined compressive strength, water stability, durability and salt swelling of the solidified soil through experiments.

Researchers have carried out a lot of experimental research and practical work on the solidification and stabilization materials of silt, and have achieved good theoretical analysis and experimental results. The solidified materials have gradually developed from traditional inorganic and organic types to inorganic-organic composite and biological improvement cured. However, there are still problems of single improvement effect and high technical difficulty in popularization and application of silt solidification in engineering application practice. And research on self-repair is relatively lacking Therefore, it is very necessary to further study the changes of the road performance of the solidified silt under the unfavorable conditions of actual work, such as water, temperature and natural or load effects, on the basis of the basic performance research on the strength and deformation of the solidified silt.

Sintered red mud is a solid industrial waste residue discharged from the production of alumina by the alkaline process. It has high calcium oxide and silicon oxide content, small particles and a network structure inside, which has strong adsorption capacity and certain hydration activity [38,39]. Asphalt is a complex mixture of hydrocarbons and their derivatives with different molecular weights. It is a temperature-sensitive material with flow self-healing properties [40–42]. Combined with the characteristics of sintered red mud particles and asphalt materials, the two materials were mixed and ground under a certain process to prepare a new type of silt curing agent, and the silt was solidified and stabilized through physical-chemical comprehensive action. The preparation process of the new curing agent can realize the value-added utilization of industrial solid waste—sintered red mud, and at the same time convert asphalt into solid powder form at normal temperature, which is beneficial to the engineering practice and construction quality control of solidified silt. On the basis of evaluating the basic mechanical properties of the composite flexible curing agent stabilized silt, the mechanical properties of stabilized silt under test conditions such as water softening, drying shrinkage, low temperature freeze-thaw, and high temperature self-healing were further studied, and the effect of the composite flexible curing agent content on water stability of stabilized silt, frost resistance and damage repair performance was analyzed. Scanning electron microscope (SEM) was used to observe the microscopic morphology of silt, cement stabilized silt and the composite flexible curing agent stabilized silt, respectively, to analyze the influence of different materials on soil structure and pore characteristics, and to explore the mechanism of solidification and stability.

## 2. Materials and Experimental

### 2.1. Materials

The soil used in the study was silt taken from the Yellow River alluvial plain in Dezhou, Shandong Province, China, located approximately 15 km away from the bank of the Yellow River. The particle analysis was conducted as shown in Figure 1. The coefficient of nonuniformity of the soil is 4.8, and its coefficient of curvature is 1.9. The basic physical index properties of the soil were given in Table 1. According to the Test Methods of Soils for Highway Engineering (JTG 3430-2021) [43], the test soil is low-liquid-limit silt containing sand and belongs to the C3 category of fine-grained soil filler.

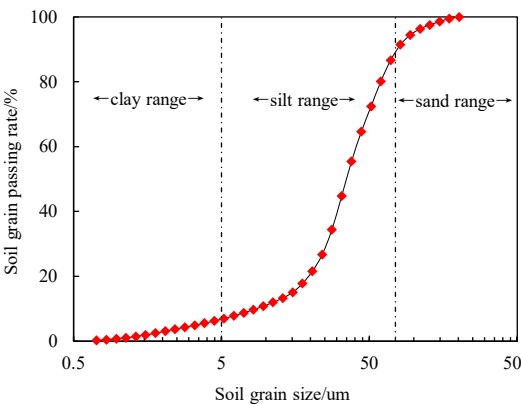

**Figure 1.** Size distribution of silt.

**Table 1.** The basic physical index properties of silt soil.

| Property | Specific Gravity | Liquid Limit/% | Plastic Limit/% | Plasticity Index | Maximum Dry Density/g·cm⁻³ | Optimum Moisture Content/% |
|---|---|---|---|---|---|---|
| Value | 2.7 | 28.4 | 19.8 | 8.6 | 1.78 | 15.1 |

The morphological characteristics of the Yellow River alluvial silt were observed by scanning electron microscope (SEM) in Figure 2. The silt has high particle roundness, uniform particle size, and the particles are connected in an overlapping manner. During rolling, it is difficult to form effective particle embedding between particles, and mutual dislocation between particles is easy to occur under the action of external force.

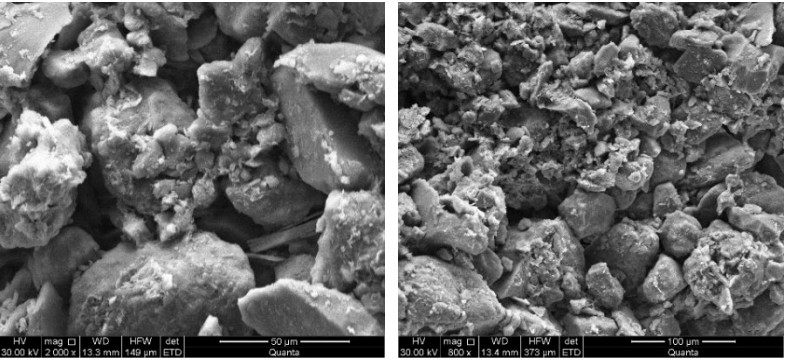

**Figure 2.** SEM images of Yellow River alluvial silt particles.

Portland cement PI42.5, Sintered red mud and base asphalt powder were used in the tested mixtures, and their properties are shown in Tables 2–4, respectively. The chemical composition of sintered red mud is mainly CaO and $SiO_2$, and it has certain hydration activity. The penetration of the base asphalt is 70 (0.1 mm), and the softening point is 46.5 °C.

**Table 2.** The properties of Portland cement.

| Test Project | Specific Gravity | Specific Surface Area m²/g | $SiO_2$% | $Al_2O_3$% | MgO% | $Fe_2O_3$% | CaO% | $Na_2O$% | $SO_3$% |
|---|---|---|---|---|---|---|---|---|---|
| Value | 3.1 | 0.35 | 23.55 | 5.64 | 1.67 | 2.85 | 64.17 | 0.26 | 0.49 |

**Table 3.** The properties of sintered red mud.

| Test Project | Water Content/% | Bulk Density g/cm$^3$ | Specific Surface Area m$^2$/kg | SiO$_2$% | Al$_2$O$_3$% | Fe$_2$O$_3$% | CaO% | Na$_2$O% |
|---|---|---|---|---|---|---|---|---|
| Value | <3 | 0.6-0.8 | 550–600 | 18.87 | 9.6 | 13.25 | 41.1 | 3.63 |

**Table 4.** The properties of base asphalt.

| Property | Density at 15 °C g/cm$^3$ | Penetration/0.1 mm | Softening Point/°C | Ductility at 15 °C/cm | Dynamic Viscosity at 60 °C/Pa·S |
|---|---|---|---|---|---|
| Value | 1.033 | 70 | 46.5 | >100 | 246 |

The composite flexible curing agent was developed with sintered red mud and No. 70 base asphalt as the main materials. First, the sintered red mud was dried and ground to less than 120 mesh for use. Then, the base asphalt was heated to a certain temperature and the ground sintered red mud was put into in proportion. After fully stirring for 120 s, it was cooled to room temperature. Finally, an appropriate amount of dispersant was put in, and the mixture was crushed to less than 0.075 mm with a pulverizer to obtain the composite flexible curing agent.

*2.2. Sample Preparation*

The samples used in the study are compacted by Proctor method according to Chinese standard Test Methods of Soils for Highway Engineering JTG 3430-2021 [43] and Test methods of materials stabilized with Inorganic Binders for Highway Engineering JTG E51-2009 [44] with 96% compaction. The formed samples are placed in a standard curing room for curing. According to engineering experience and previous studies [45], the cement content is limited to 5%. The stabilized soil scheme was given in Table 5. Here, the reference group F-0 denotes the stabilized silt with 5% cement. F-2, F-4, F-6 and F-8 denotes the compound stabilized silt using 2%, 4%, 6%, 8% flexible curing agent on the basis of 5% cement.

**Table 5.** The stabilized soil scheme.

| Sample | Cement/% | Flexible Curing Agent/% |
|---|---|---|
| F-0 | 5 | 0 |
| F-2 | 5 | 2 |
| F-4 | 5 | 4 |
| F-6 | 5 | 6 |
| F-8 | 5 | 8 |

*2.3. Test Methods*

2.3.1. Compressive Strength

The compressive strength of soil specimens (Φ39.1 mm × 80 mm) are tested at 3, 7 and 28 standard curing ages using a compression testing machine according to the Chinese standard JTG E51 T0805(2009) [44]. The average value of at least three specimens is reported as the compressive strength test result of the specimen group.

Another set of specimens are prepared to soak in water for one day at the last day of 3, 7 and 28 standard curing ages. The compressive strength after immersion are tested.

2.3.2. Drying Shrinkage

The drying shrinkage are measured in accordance with JTG E51 T0854 (2009) [44]. All drying shrinkage samples (100 mm × 100 mm × 400 mm) are placed in a room with 20 ± 2 °C and 60 ± 5% relative humidity after standard curing for 6 d and soaking in water for 1 d. Measurements are carried out until 30 days drying period.

### 2.3.3. Low Temperature Freeze-Thaw

The low temperature freeze-thaw of soil specimens (Φ150 mm × 150 mm) are tested at 28 d. according to the Chinese standard JTG E51 T0858 (2009) [44]. The samples are first placed in a room with −18 °C for 16 h, and then melt in 20 °C water tank for 8 h after freezing. The freeze -thaw cycles are performed 5 times.

### 2.3.4. High Temperature Self-Healing

The high temperature self-healing samples (Φ150 mm × 150 mm) are first loaded to 0.9 times the maximum load, and then heated to 49 ± 1 °C after sealing. Finally, it returned to 20 °C after keeping for 2 h at 49 ± 1 °C. The heating cycles are performed 5 times.

### 2.3.5. Scanning Electron Microscopy (SEM)

The morphology of soil specimens are analyzed by Field Emission Scanning Electronic Microscopy (Sigma 500, Carl Zeiss AG, Oberkochen, Germany). The microstructural differences of silt, cement stabilized silt, the flexible curing agent stabilized silt and the flexible curing agent stabilized silt after heating are observed.

## 3. Results and Discussion

### 3.1. Compressive Strength and Water Stability

Figure 3 shows the effect of the flexible curing agent on the compressive strength of silt at 3, 7 and 28 days. The compressive strengths increase with ages in all series. The compressive strengths at 3 and 7 days are increased rapidly, but tend to be stable at 28 days. The addition of the flexible curing agent has a significant effect on the early strength of stabilized silt. Figure 4 shows the normalized compressive strength of all stabilized soil compared to 0%. The results show that the ternary cementitious system containing the flexible curing agent and cement in silt soil behaves obviously better than the effect of cement alone. The compressive strength of F-0 is 0.80 and 0.91 MPa at the 7 and 28 days, respectively (Table 6). The addition of 2%, 4%, 6% and 8% of the flexible curing agent cause increase respectively about 17.5%, 56.3%, 67.5%, 61.3% at 7 days, and 11.0%, 44.0%, 58.2%, 50.5% at 28 days. As demonstrated in Figure 4, the highest compressive strength rates of stabilized silt are achieved when F-6 blend is added. This incorporation led to approximately 58%–68% of strength increase higher than that of F-0 at 7 and 28 days. The compressive strength of F-8 decreases compared with that of F-6. This phenomenon is attributed to plasticity enhancement with the increase of asphalt content.

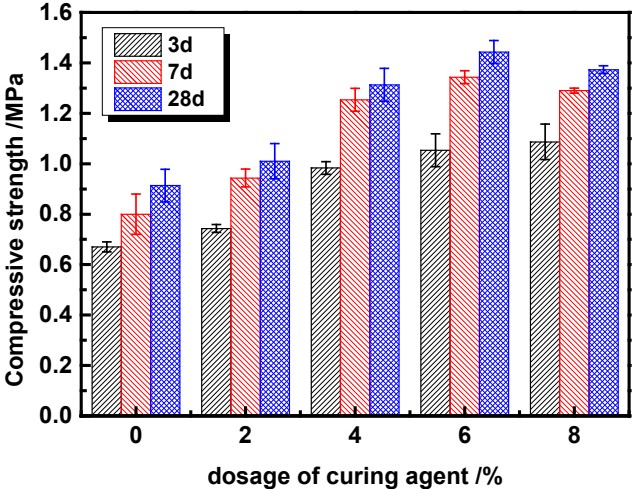

**Figure 3.** Effect of dosage of the flexible curing agent on compressive strength of stabilized silt with curing ages.

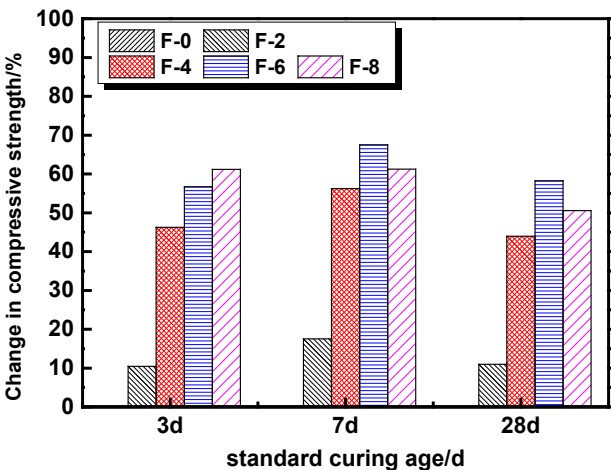

**Figure 4.** Normalized compressive strength of all stabilized soil related to F-0.

In order to investigate the water stability of the flexible curing agent stabilized silt, the water immersion strength test was carried out. Table 6 shows the strength change before and after immersion for all series at 3, 7 and 28 days. After immersion in water, the compressive strengths of silt are decreased compared with standard curing. This result is consistent with those observed previously by other researcher [46,47]. This is attributed to that the ingress of water damages the internal structure of the solidified soil, the gel material is peeled off from the soil particles, and the number of pores in the sample increases. However, the addition of the flexible curing agent reduces the strength loss rate significantly, as shown in Figure 5. The strength loss rate of F-0 is 61.2%, 51.3% and 50.5% at 3, 7 and 28 days, respectively. However, the addition of 2%, 4%, 6% and 8% of the flexible curing agent cause reduction in strength loss rate respectively about 35.9%, 65.0%, 73.5%, 68.3% at 3 days, about 52.2%, 76.6%, 81.1%, 80.3% at 7 days, and about 60.8%, 85.0%, 86.3%, 81.2% at 28 days. The lowest compressive strength loss rates of stabilized silt are achieved when 6% flexible curing agent blend is added. This incorporation led to approximately 81-86% of strength increase higher than that of F-0 at 7 and 28 days. And the strength loss rate of F-8 increases slightly compared to that of F-6, which is similar to the result in standard compressive strength. As can be seen in Figure 5, after 7 days of curing age, the change range of immersion strength loss rate is small in all series, indicating that prolonging the curing time is conducive to enhancing the water stability of the stabilized silt. Moreover, when the flexible curing agent dosage is higher than 4%, the immersion strength loss rates of stabilized silt gradually tend to be stable.

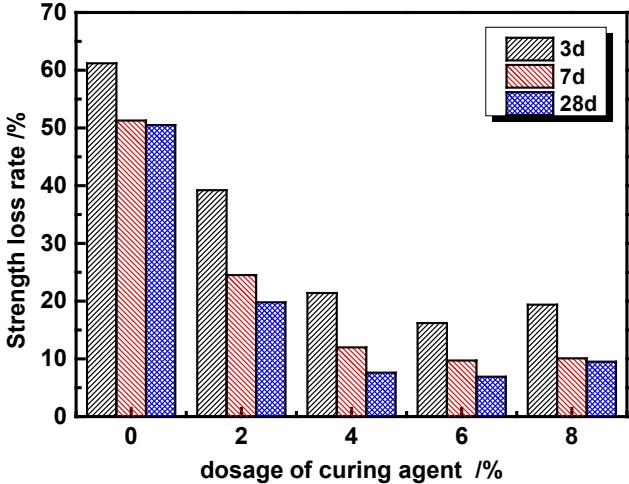

**Figure 5.** Strength loss rate of stabilized silt after immersion.

**Table 6.** Standard and immersion compressive strength values of all specimens.

| Sample | Compressive Strength/MPa | | | | | | | | | | | |
| | 3 d | | | | 7 d | | | | 28 d | | | |
| | Standard Curing | Standard Deviation | Immersion | Standard Deviation | Standard Curing | Standard Deviation | Immersion | Standard Deviation | Standard Curing | Standard Deviation | Immersion | Standard Deviation |
|---|---|---|---|---|---|---|---|---|---|---|---|---|
| F-0 | 0.67 | 0.020 | 0.26 | 0.022 | 0.80 | 0.080 | 0.39 | 0.050 | 0.91 | 0.065 | 0.45 | 0.070 |
| F-2 | 0.74 | 0.015 | 0.45 | 0.030 | 0.94 | 0.035 | 0.71 | 0.055 | 1.01 | 0.070 | 0.81 | 0.045 |
| F-4 | 0.98 | 0.025 | 0.77 | 0.020 | 1.25 | 0.045 | 1.10 | 0.025 | 1.31 | 0.065 | 1.21 | 0.015 |
| F-6 | 1.05 | 0.065 | 0.88 | 0.070 | 1.34 | 0.025 | 1.21 | 0.05 | 1.44 | 0.045 | 1.34 | 0.020 |
| F-8 | 1.08 | 0.070 | 0.87 | 0.055 | 1.29 | 0.01 | 1.16 | 0.01 | 1.37 | 0.015 | 1.24 | 0.060 |

This phenomenon with higher compressive strengths and water stabilities of the flexible curing agent stabilized silt series is caused for two reasons. One is attributed to the sintered red mud in the flexible curing agent. Sintered red mud is an active component and can stimulate the cement hydration reaction. The other is attributed to the asphalt that works in the soil. The asphalt particles connect the hydration products and the silt particles to form a cohesive skeleton structure under the action of physical compaction.

### 3.2. Drying Shrinkage

The effects of the flexible curing agent on drying shrinkage of stabilized silt are illustrated in Figures 6–9. Figure 6 shows the variation of loss rate with time. The change of water loss rate with time for silt soil is very different from that of stabilized soil. On the first day of the test, the water loss rate of silt soil reaches 11.65%, accounting for 88% of its final water loss rate at 12 days. As to stabilized silt, the water loss rate changes the most in 7 days. The water loss rate of F-0 is 14.9% at 7 test days, accounting for 89.9% of its final water loss rate. The water loss rate of F-2, F-4 and F-6 is 12.2%, 11.9% and 11.0% respectively at 7 days, and accounting for 87.5%, 86.6% and 85.5% of their final water loss rate at 28 days. The result shows that the addition of cement increases the water loss of silt soil, but the addition of the flexible curing agent decreases the water loss of cement stabilized silt. Compared to the silt soil, all the flexible curing agent series show better water retention in the early stage, which can inhibit the occurrence of shrinkage cracks in the specimen to a certain extent. It is recommended to take maintenance measures within 7 days after the completion of construction to avoid excessive water loss and increase dry shrinkage strain.

Figure 7 exhibits the shrinkage strain variation of silt and stabilized soil with time. The drying shrinkage of silt develops quickly and diverges from the flexible curing agent stabilized silt at early test ages. The dry shrinkage strain of silt soil mainly occurred in the first day of the dry shrinkage test. the shrinkage strain of silt soil reaches $1302.2 \times 10^{-6}$ $\mu\varepsilon$, accounting for 89.5% of its total strain at 12 days. In Figure 7, we observe that the dry shrinkage strain of stabilized silt mainly occurred in the first 14 days, and the flexible curing agent series demonstrate lower drying shrinkage value than silt and cement stabilized silt at early age. The shrinkage strain of F-0 is $1130.8 \times 10^{-6}$ $\mu\varepsilon$ at 14 test days, accounting for 89.3% of its total strain. The shrinkage strain of F-2, F-4 and F-6 is $634.3 \times 10^{-6}$ $\mu\varepsilon$, $535.6 \times 10^{-6}$ $\mu\varepsilon$ and $443.8 \times 10^{-6}$ $\mu\varepsilon$ respectively at 14 days, and accounting for 87.3%, 85.4% and 84.4% of their total strain. The incorporation of the flexible curing agent can effectively decrease the shrinkage strain of silt. And the best performance is obtained by the addition 6% curing agent whose drying shrinkage observably decreased and was lowest.

Figure 7 exhibits the shrinkage coefficient variation of silt and stabilized soil with time. With the increase of time, the drying shrinkage coefficient shows a change rule that increases first and then stabilizes for all series. The average shrinkage coefficient of silt, F-0, F-2, F-4 and F-6 is $183.2 \times 10^{-6} \cdot °C^{-1}$, $172.1 \times 10^{-6} \cdot °C^{-1}$, $171.9 \times 10^{-6} \cdot °C^{-1}$, $134.5 \times 10^{-6} \cdot °C^{-1}$ and $122.7 \times 10^{-6} \cdot °C^{-1}$, respectively. The incorporation of 4%–6% curing agent led to approximately 22%–29% of average shrinkage coefficient decrease than that of F-0. It shows that the stabilized silt with the flexible curing agent has better shrinkage resistance.

It is obvious that the drying shrinkage of the material is closely related to the loss of moisture. Figure 9 shows the relationship between water loss rate and dry shrinkage strain of silt and stabilized silt. The water loss rate of stabilized silt in the early stage has little effect on its dry shrinkage strain, but it is larger in the later stage. When the water loss rates of stabilized silt are less than 10%, their dry shrinkage strain hardly increase, but when the water loss rates exceed 10%, the dry shrinkage strains increase rapidly. The loss of moisture in the early stage is mainly the surface moisture of the test specimens and the free water inside the specimens, the loss of that has little effect on the internal pore structure of specimens. In the later stage, the capillary water and adsorbed water inside the specimens are lost, making the water film on the surface of the soil particles thinner, the spacing between the particles smaller, and the molecular force increased. The loss of capillary water and adsorbed water inside the specimens is the main cause of dry shrinkage.

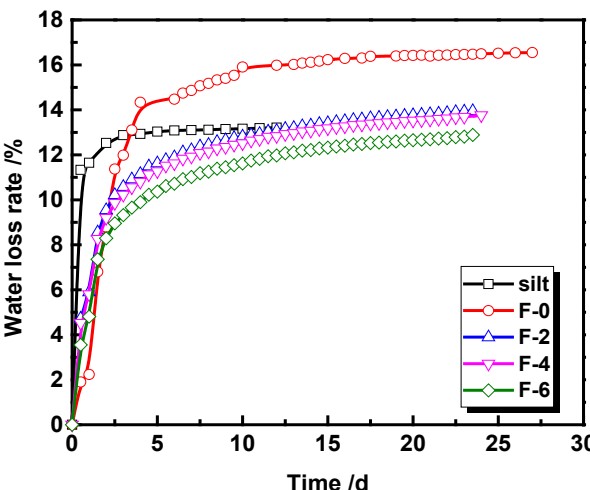

**Figure 6.** Variation of water loss rate with time.

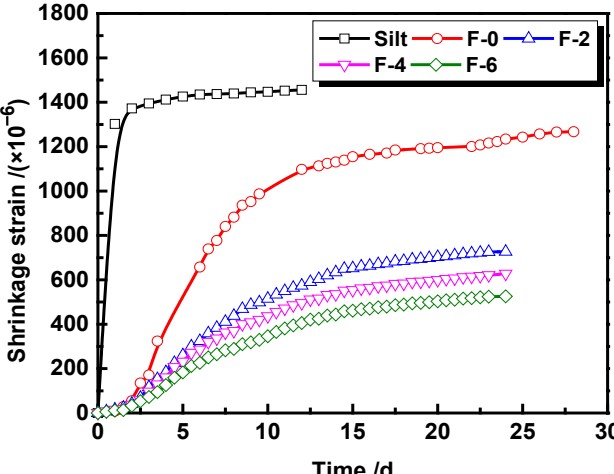

**Figure 7.** Variation of shrinkage strain with time.

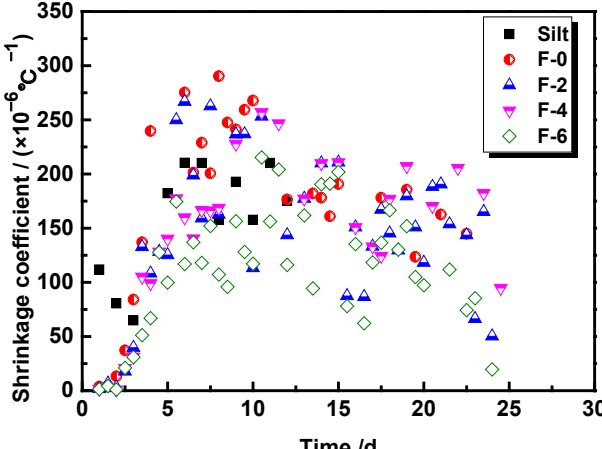

**Figure 8.** Variation of shrinkage coefficient with time.

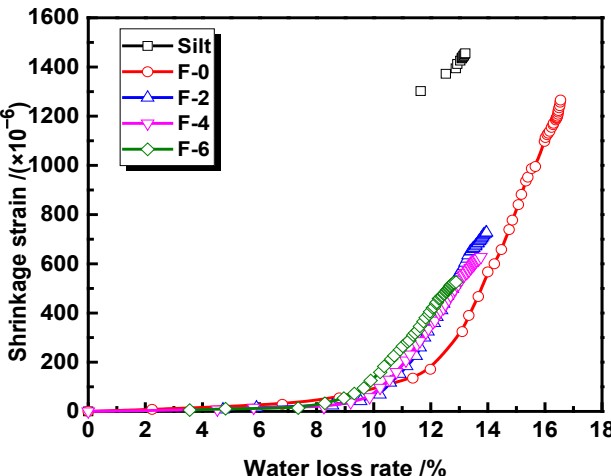

**Figure 9.** Relationship between water loss rate and dry shrinkage strain.

*3.3. Low Temperature Freeze-Thaw*

The mechanical behavior of the flexible curing agent stabilized silt under low temperature freeze-thaw was analyzed. The crack and damage occurred in F-0 after only 1 freeze- thaw cycle (Figure 10), so 2, 4, 6, 8% curing agent stabilized silt was selected for low-temperature freeze-thaw. The stress-strain curves of curing agent stabilized silt under standard curing and low temperature freeze-thaw are compared in Figure 11. The results indicate that the strength and stiffness of curing agent stabilized silt all decrease after 5 freeze-thaw cycles. The strength change rate of F-2, F-4 and F-6 is −20.5%, −12.9%, −10.1%, and −14.6% respectively as seen in Figure 12. Compared to the cement stabilized silt (F-0), the incorporation of the flexible curing agent improves the freeze resistance ability. The best performance of freeze resistance is obtained by the addition 6% curing agent whose strength loss observably decreased and the strength loss rate was lowest. This is contributed to that the active components of sintered red mud in the flexible curing agent component stimulate the cement hydration reaction, and the asphalt particles in the flexible curing agent blocks the capillary connection in the structure through granulation dispersion and viscous adsorption, reducing the porosity. However, the strength loss increased by the addition of 8% curing agent. This difference is caused by the overdose of the flexible curing agent which makes the form of agglomerate structure and microporosity, increases water absorption and decreases the freeze resistance ability.

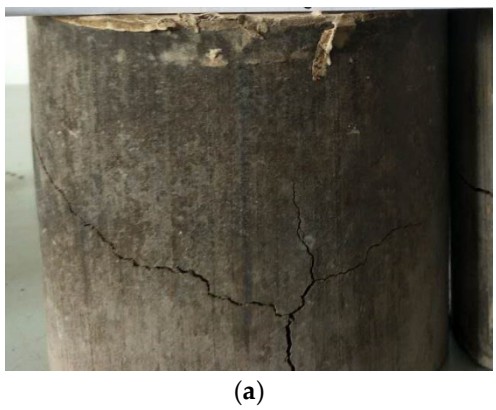
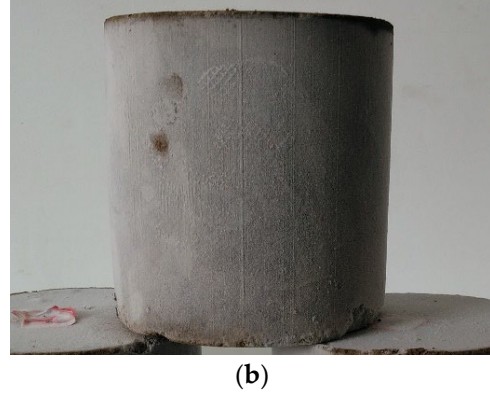

**(a)**                                                                 **(b)**

**Figure 10.** Soil sample state after low temperature freeze-thaw. (**a**) Cement stabilized; (**b**) flexible curing agent stabilized.

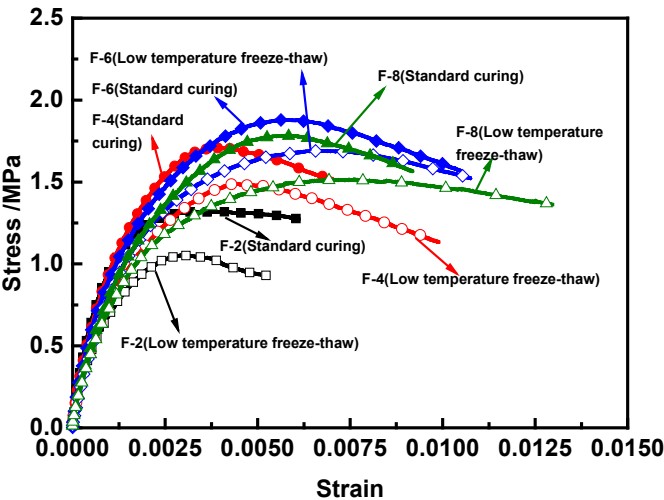

**Figure 11.** Stress-strain curve of stabilized silt under standard curing and low temperature freeze-thaw.

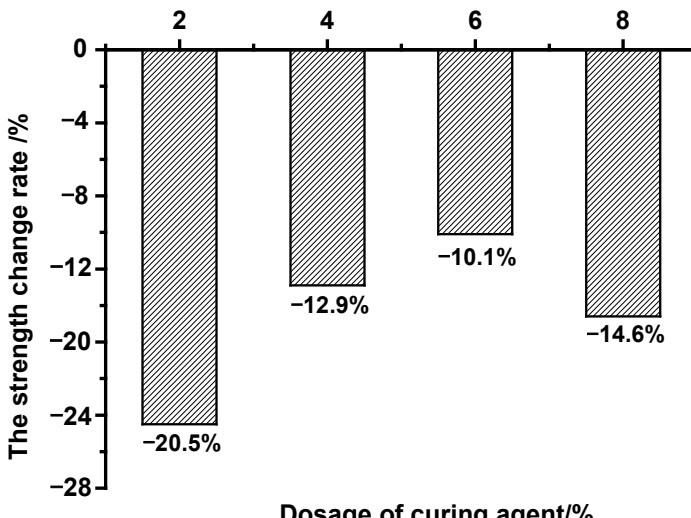

**Figure 12.** The strength change rate of the flexible curing agent stabilized silt.

### 3.4. High Temperature Self-Healing

The damaged asphalt material can be reorganized and self-healed under high temperature conditions [40,42]. The high temperature self-healing test was conducted to explore the effect of asphalt self-healing on soil remediation. The stress-strain curve of the flexible curing agent stabilized silt under standard curing and high temperature are compared in Figure 13. It can be seen that the slopes of the σ-ε curves of the high-temperature self-healing test are all smaller than those of the standard curing σ-ε curve, and the peak values of the curve gradually increase beyond those of the standard curing curve. The results indicate that the compressive modulus of the stable silt decreases in the high temperature healing test, and the compressive strength is enhanced with the increase of the flexible curing agent content. The strength change rate of F-2, F-4 and F-6 is −8.3%, −2.3%, +8.0%, and +12.9% respectively as seen in Figure 14. The self-healing properties of asphalt components play a role. The stabilized silt with 6%–8% curing agent shows a good self-healing and reinforcing effect. The interface surface energy of asphalt and soil particles is changed significantly under high temperature, where the diffusion and reorganization of asphalt molecules are stimulated. The reconstituted asphalt fill and adhere to the micro-cracks in the soil, which enhance compactness and improve the microstructure in damaged stabilized soil. There is a decrease in modulus with the addition of the flexible curing agent. The modulus of F-2, F-4 and F-6 is 1321 MPa, 1118 MPa, 1013 MPa and 907 MPa respectively.

This result is attributed to that the increase of asphalt content enhances the plasticity of the soil.

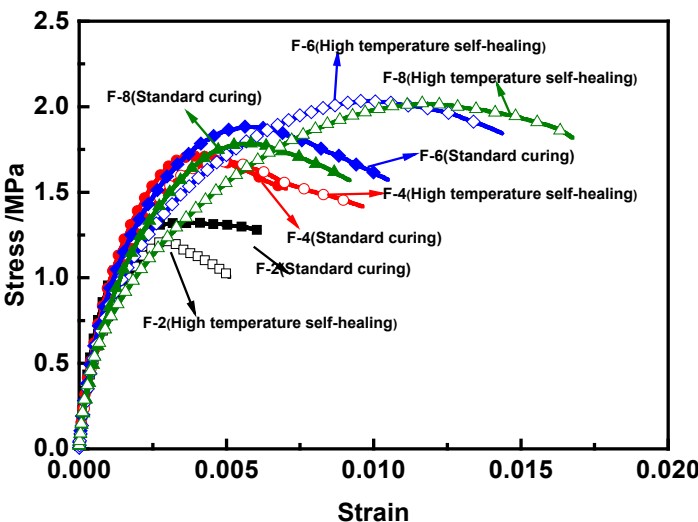

**Figure 13.** Stress-strain curve of the flexible curing agent stabilized silt under standard curing and high temperature.

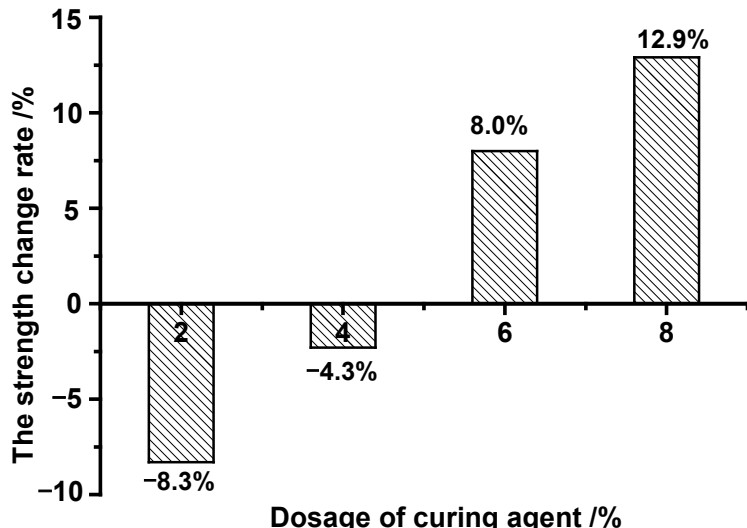

**Figure 14.** The strength change rate of the flexible curing agent stabilized silt.

*3.5. Microstructure*

The microstructures of silt, cement stabilized silt, the flexible curing agent stabilized silt, the flexible curing agent stabilized silt after heating at 28 days are shown in Figure 15. In considering the microstructure of silt, high particle roundness and overlapping connected manner are observed between particles. The clay content in the soil is small, there is no obvious bonding between particles, and the structure is loose and granular, and the pores between the particles are large. During rolling, it is difficult to form effective particle embedding between particles, and mutual dislocation between particles is easy to occur under the action of external force. A lot of flocculated calcium silicate hydrate gel and acicular ettringite are found on the surface of cement-stabilized silt particles in Figure 15b, forming a network structure. Despite this, the bonding between soil particles and hydration products is not tight, and there are more connected intergranular pores, resulting in low structural density. In the flexible curing agent stabilized silt, the asphalt component has obvious bonding effect with hydration products and soil particles, as shown

in Figure 15c,d. It forms many disconnected pores uniformly distributed non-connected micropore structures, which improve the microstructure development of silt.

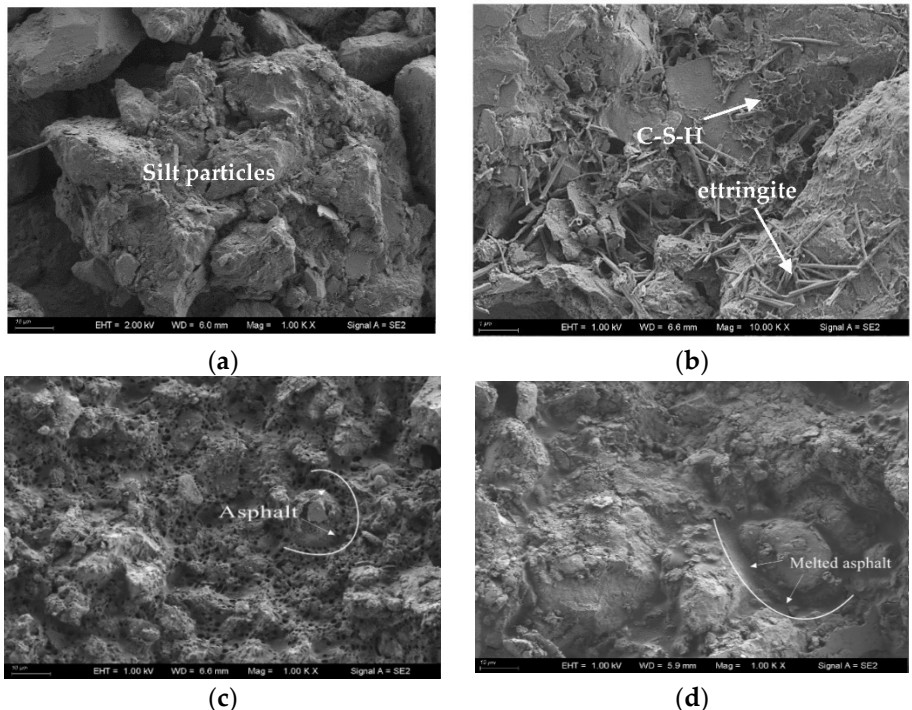

**Figure 15.** SEM morphology photographs of different samples. (**a**) Silt; (**b**) Cement stabilized silt; (**c**) Flexible curing agent stabilized silt; (**d**) Flexible curing agent stabilized silt after heating.

## 4. Conclusions

Based on the limited testing results, the following conclusion can be drawn.

(1) The inclusion of the flexible curing agent has a significant contribution in improving the compressive strength, water stability and resistance ability to drying shrinkage and freeze-thaw because of cementing effect of the cement hydrates and asphalt. For the ternary cementitious system containing the flexible curing agent and cement in silt soil, the system behaves well in improving the mechanical properties, better than the effect of cement alone. Moreover, it has good high temperature self-healing ability for the addition of asphalt, which will play an important role in the recovery of soil damage.

(2) Asphalt is a viscoelastic plastic material. An appropriate amount of asphalt can make the soil obtain a stable asphalt cementitious structure and improve the performance of the soil together with the cement hydration products. However, the increase of the asphalt content will enhance the plasticity of the soil and reduce the mechanical properties such as strength, water stability and resistance to low temperature crack. The preferred dosage 4%~6% of the flexible curing agent is obtained by mechanical property analysis.

(3) The SEM images show that the incorporation of the flexible curing agent helps the silt form dense cementation and non-connected microporous structure, that is beneficial to the improvement of water stability and frost resistance. The asphalt component can reorganize and diffuse, filling internal pores and micro-cracks, and achieving soil damage repair and structural reinforcement.

Based on the actual application conditions, the mechanical properties evolution test and analysis of the flexible solidified silt under indoor simulation conditions are carried out in this paper. Next, related research work on dynamic loading and fatigue performance under traffic load will be carried out in combination with the application horizon of solidified silt.

**Author Contributions:** Conceptualization, L.W. and W.W.; methodology, Z.S. (Zhiyi Sai); software, W.W.; validation, Z.S. (Zhiyi Sai), H.H. and G.H.; formal analysis, H.W.; investigation, L.Z.; resources, L.W.; data curation, J.W.; writing—original draft preparation, W.W.; writing—review and editing, W.W.; visualization, J.W.; supervision, J.W.; project administration, Z.S. (Zhaoyun Sun); funding acquisition, G.H. All authors have read and agreed to the published version of the manuscript.

**Funding:** This research was funded by the National Key R&D Program of China, grant No. 2018YFB1600103, the National Natural Science Foundation of China, grant No. 42107213, Shandong Provincial Natural Science and Foundation, grant No. ZR2020QE271 and Shandong Provincial Key Research and Development Program, grant No. 2019GSF109020.

**Institutional Review Board Statement:** Not applicable.

**Informed Consent Statement:** Not applicable.

**Data Availability Statement:** Not applicable.

**Conflicts of Interest:** The authors declare no conflict of interest.

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
