# Peer review of "Mechanical and Self-Healing Performance of Yellow River Alluvial Silt Treated with Composite Flexible Curing Agent"

_coatings, doi:10.3390/coatings12060870_

Round 1
Reviewer 1 Report
This article investigated stabilized Yellow River alluvial silt's mechanical properties with a flexible composite curing agent. The article requires some revision and the comments are listed below.
- Line 1 “This report presents” I suggest using the sentences “This article or research presents”.
- Abstract, the description of the work investigated is unclear and should be improved with more detail.
- Most of the literatures were cited in a group. However, more individual literatures regarding compressive strength, drying shrinkage, freeze-thaw and high temperature should be discussed in the introduction section.
- Line 149, Add the properties of Portland cement, sintered mud and asphalt powder.
- Line 163, Why is the cement content limited to 5%? Highlight this in the manuscript.
- Line 163, on what basis the 2%, 4%, 6%, 8% flexible curing agents were selected? Why is it limited to 8%?
- Table 2, why is the standard curing compressive strength higher than the immersion curing?. The reason should be discussed in section 3.1.
- Compressive strength results, section 3.2-3.4 results can be compared with earlier findings. Results correlation can further improve the reliability of the investigation.
- Figure 7, calcium silicate hydrate gel, flaky calcium hydroxide and acicular ettringite should be marked or shown in the figure for better understanding.
- Conclusions should be improved. Quantify the results and include the key findings.
Author Response
1- Line 1 “This report presents” I suggest using the sentences “This article or research presents”.
Response: Thank you for your suggestion, we have made corresponding modification in red words.
2-Abstract, the description of the work investigated is unclear and should be improved with more detail.
Response: Thank you for your suggestion, we have improved the abstract and made corresponding modification in red words.
3-Most of the literatures were cited in a group. However, more individual literatures regarding compressive strength, drying shrinkage, freeze-thaw and high temperature should be discussed in the introduction section.
Response: Thank you for your suggestion, we have improved the introduction section and made corresponding modification in red words.
4-Line 149, Add the properties of Portland cement, sintered mud and asphalt powder.
Response: Thank you for your suggestion, the properties of Portland cement, sintered mud and asphalt powder have been added in table 2-4 and corresponding modification has been made in red words.
5-Line 163, Why is the cement content limited to 5%? Highlight this in the manuscript.
Response: We are very sorry for our unclear report in designing of the cement content. The design of the cement content is based on the engineering experience and previous study results. In China, the cement content in the cement-solidified soil for engineering is mostly 4%-6%, the cement content of 5% is selected as the reference group for the experimental comparison in this paper.
Corresponding modifications have been made in the revised manuscript in red words in 2.2 according to the reviewer’s comments.
6-Line 163, on what basis the 2%, 4%, 6%, 8% flexible curing agents were selected? Why is it limited to 8%?
Response: The design results are based on a large number of tests. Initially, based on the research experience and some literatures on solidified soil, 5%,10% and 15% flexible curing agents was selected in initial research and the compressive strength was taken as the evaluation parameter of stabilization effect. The results showed that the strength of the solidified silt with 10% content decreased significantly compared with that of 5% content. Then, the 2%, 4%, 6%, 8%,10% flexible curing agents were selected in next research. The addition of 8% flexible curing agents causes decrease in compressive strength compared with that of 6% flexible curing agents. This phenomenon is attributed to plasticity enhancement with the increase of asphalt content. Therefore, it is limited to 8% flexible curing agents in this paper considering the influence of asphalt content.
7-Table 2, why is the standard curing compressive strength higher than the immersion curing?. The reason should be discussed in section 3.1.
Response: The standard curing compressive strength is higher than the immersion curing. Some researchers also reported the same conclusion (Fan, 2019 and 2021).
The composite flexible curing agent was prepared by sintering red mud and asphalt as the main raw materials. The sintered red mud contains active CaO and SiO2 and has hydration activity. Under standard curing conditions, the cement and red mud in curing agent undergoes hydration reaction to form C-S-H and other products. Moreover, the asphalt particles connect the hydration products and silt particles under the action of physical compaction. However, when the sample is soaked in water, the internal structure of the solidified silt is destroyed, the gel material is peeled off from the soil particles, and the number of pores in the sample increases. As a result, the compressive strength of sample under standard curing condition is higher than that of sample under the immersion curing condition.
Some improvements were made in the article and the reference was added accordingly.
Liang Fan, Shanwei Mou,Yong zhen Li. Strength and Water Stability of Compound Stabilized Silt with Emulsified Asphalt. Highway Engineering, 2019,44(4): 178-238
Liang Fan, Sheng-jie Zhou, Jia-lin Hou, Lin Wang. Composite Stabilization of Silty Soil near the Yellow River: Two Methods and Performance Comparison. Journal of Yangtze River Scientific Research Institute, 2021,38(12):118-124.
8-Compressive strength results, section 3.2-3.4 results can be compared with earlier findings. Results correlation can further improve the reliability of the investigation.
Response: Thank you for your suggestion. We have made corresponding modification in section 3.2-3.4, and the earlier findings was compared.
9-Figure 7, calcium silicate hydrate gel, flaky calcium hydroxide and acicular ettringite should be marked or shown in the figure for better understanding.
Response: Thank you for your suggestion. Calcium silicate hydrate gel, flaky calcium hydroxide and acicular ettringite are marked in Figure.7.
10-Conclusions should be improved. Quantify the results and include the key findings.
Response: Thank you for your suggestion. The conclusions has been rewritten to be more concise.
Reviewer 2 Report
The manuscript entitled " Mechanical Behavior of Yellow River Alluvial Silt Treated with Composite Flexible Curing Agent” presented study of curing agent.. The influence of different parameters was studied and analyzed. The manuscript lacks clarity and needs much improvement before further processing. It seems like this paper is directly made from thesis without putting the additional efforts required for writing manuscripts.
This reviewer recommends minor editing and resubmits for re-review.
Comments:
- The English writing of the manuscript needs improvement. Therefore, it could benefit greatly from professional editing to improve technical writing and English.
- Please mention your study limits and suggest some future research topics
- In References, the sources are written in different styles. Please update the reference list. It is necessary to bring in accordance with the requirements of the magazine for the design of References. If possible, indicate DOI.
- Please use some innovative keywords.
- Please mention your study limits in the abstract.
- The Conclusions should reflect what the practical application of the results obtained in this study is. In what climatic conditions should the recommendations of the authors be taken into account?
- The authors should increase their discussion on previous related research and highlight how their study is providing a different approach or adding significantly to what has been done. The authors have to explain what is the new here in comparison with the previous studies. The novelty of the current work should be highlighted in the introduction. Please try to mention a problem that needs solving - in other words, the research question underlying your study clearer.
- The title of the manuscript should be revised.
- Some types of standards should be used to perform different experimental studies. Please provide details for the standards used in each study.
- Section 4 should be discussed in detail.
- The authors must redo the Abstract and bring it in compliance with the requirements of the journal. The scientific problem is poorly described (Background). The scientific novelty is not indicated. I recommend shortening the Abstract to 200 words. Editors strongly encourage authors to use the following style of structured abstracts, but without headings: (1) Background: Place the question addressed in a broad context and highlight the purpose of the study; (2) Methods: Briefly describe the main methods or treatments applied; (3) Results: Summarize the article's main findings; and (4) Conclusions: Indicate the main conclusions or interpretations. The abstract should be an objective representation of the article
- It is advisable to add a flowchart at the beginning of the paper. Then the article would become more visual and structured
- Figure 6 can be replaced with column bar chart.
- The economic aspects are also required for sustainability in social aspect. It is suggested to authors to evaluate the cost-benefit study of this as a further investigation
- The conclusion should be an objective summary of the most important findings in response to the specific research question or hypothesis. A good conclusion states the principal topic, key arguments and counterpoint, and might suggest future research. It is important to understand the methodological robustness of your study design and report your findings accordingly. Please improve your conclusion section.
Author Response
Special thanks to you for your good comments. We have carefully revised the manuscript, and the paper was also re-scrutinized to improve the English. Our responses are as follows, and we amended the relevant part in revised manuscript and the changes below are noted in red words in the article.
1-The English writing of the manuscript needs improvement. Therefore, it could benefit greatly from professional editing to improve technical writing and English.
Response: Special thanks to you for your good suggestion. We have carefully revised the manuscript, and the paper was re-scrutinized to improve the English.
2-Please mention your study limits and suggest some future research topics.
Response: Thank you for the suggestion. Based on the actual application conditions, the mechanical properties evolution test and analysis of the flexible solidified silt under indoor simulation conditions are carried out in this paper. Next, related research work on dynamic loading and fatigue performance under traffic load will be carried out in combination with the application horizon of solidified silt. We have added the study limits and future research topics at the ending of the conclusions.
3-In References, the sources are written in different styles. Please update the reference list. It is necessary to bring in accordance with the requirements of the magazine for the design of References. If possible, indicate DOI.
Response: Thank you for the suggestion. The refence list has been updated according to the requirements of the magazine.
4-Please use some innovative keywords.
Response: Thank you for the suggestion. The keywords have been revised in the manuscript.
5-Please mention your study limits in the abstract.
Response: Thank you for the suggestion. we have improved the abstract and made corresponding modification in red words. The study limits was added in the conclusion section.
6-The Conclusions should reflect what the practical application of the results obtained in this study is. In what climatic conditions should the recommendations of the authors be taken into account?
Response: Special thanks for your suggestion and question. Based on a large amount of data research, combined with the climate zone where the silt is located, the conditions of the high temperature repair test are limited in this paper. Due to the complexity of climatic conditions in practical applications and the applicability of other soil source types have not been fully studied, the influence of climatic conditions has not been discussed in this paper.
7-The authors should increase their discussion on previous related research and highlight how their study is providing a different approach or adding significantly to what has been done. The authors have to explain what is the new here in comparison with the previous studies. The novelty of the current work should be highlighted in the introduction. Please try to mention a problem that needs solving - in other words, the research question underlying your study clearer.
Response: Special thanks to your suggestion. Previous related research was added in the corresponding part. The novelty of our work was also highlighted in the introduction. The introduction section has been revised according to your suggestion.
Researchers have carried out a lot of experimental research and practical work on the solidification and stabilization materials of silt, and have achieved good theoretical analysis and experimental results. The solidified materials have gradually developed from traditional inorganic and organic types to inorganic-organic composite and biological improvement cured. However, there are still problems of single improvement effect and high technical difficulty in popularization and application of silt solidification in engineering application practice. And research on self-repair is relatively lacking Therefore, it is very necessary to further study the changes of the road performance of the solidified silt under the unfavorable conditions of actual work, such as water, temperature and natural or load effects, on the basis of the basic performance research on the strength and deformation of the solidified silt.
Sintered red mud is a solid industrial waste residue discharged from the production of alumina by the alkaline process. It has high calcium oxide and silicon oxide content, small particles and a network structure inside, which has strong adsorption capacity and certain hydration activity. Asphalt is a complex mixture of hydrocarbons and their derivatives with different molecular weights. It is a temperature-sensitive material with flow self-healing properties. Combined with the characteristics of sintered red mud particles and asphalt materials, the two materials were mixed and ground under a certain process to prepare a new type of silt curing agent, and the silt was solidified and stabilized through physical-chemical comprehensive action. The preparation process of the new curing agent can realize the value-added utilization of industrial solid waste - sintered red mud, and at the same time convert asphalt into solid powder form at normal temperature, which is beneficial to the engineering practice and construction quality control of solidified silt.
On the basis of evaluating the basic mechanical properties of the composite flexible curing agent stabilized silt, the mechanical properties of stabilized silt under test conditions such as water softening, drying shrinkage, low temperature freeze-thaw, and high temperature self-healing were further studied, and the effect of the composite flexible curing agent content on water stability of stabilized silt, frost resistance and damage repair performance was analyzed. Scanning electron microscope (SEM) was used to observe the microscopic morphology of silt, cement stabilized silt and the composite flexible curing agent stabilized silt, respectively, to analyze the influence of different materials on soil structure and pore characteristics, and to explore the mechanism of solidification and stability.
8-The title of the manuscript should be revised.
Response: Thank you for the suggestion. The title of the manuscript has been revised as “Mechanical and Self-healing Performance of Yellow River Alluvial Silt Treated with Composite Flexible Curing Agent”
9-Some types of standards should be used to perform different experimental studies. Please provide details for the standards used in each study.
Response: Special thanks for your suggestion. The Chinese standards, JTG E51 and JTG 3430 were used in the study. JTG E51 refers to Test methods of materials stabilized with Inorganic Binders for Highway Engineering, and JTG 3430 refers to Test Methods of Soils for Highway Engineering. The details of the standards were added in corresponding part in red words.
10-Section 4 should be discussed in detail.
Response: Thank you for your suggestion. The discussion details are shown in Section 3, and this part has been improved.
11-The authors must redo the Abstract and bring it in compliance with the requirements of the journal. The scientific problem is poorly described (Background). The scientific novelty is not indicated. I recommend shortening the Abstract to 200 words. Editors strongly encourage authors to use the following style of structured abstracts, but without headings: (1) Background: Place the question addressed in a broad context and highlight the purpose of the study; (2) Methods: Briefly describe the main methods or treatments applied; (3) Results: Summarize the article's main findings; and (4) Conclusions: Indicate the main conclusions or interpretations. The abstract should be an objective representation of the article
Response: Thank you for the suggestion. we have improved the abstract and made corresponding modification in red words.
12-It is advisable to add a flowchart at the beginning of the paper. Then the article would become more visual and structured
Response: Special thanks for your suggestion, and we have also considered your suggestion seriously. However, considering that the content of the test is not complicated, only a few mechanical property tests and SEM analysis, we think it is not appropriate to add the flow chart to the text, Therefore, no flowchart is added in the text.
13-Figure 6 can be replaced with column bar chart.
Response: Thank you for the suggestion. Figure 6 shows the Variation of water loss rate with time in drying shrinkage test. It is not suitable to use a bar chart instead. We're guessing you're referring to Figure 5 rather than Figure 6, and we've modified it accordingly.
14-The economic aspects are also required for sustainability in social aspect. It is suggested to authors to evaluate the cost-benefit study of this as a further investigation
Response: Thank you for the suggestion. The engineering construction in China is faced with the serious shortage of earth and rock resources. The silt treated by the technique in this paper can replace the cement stabilized crushed stone material as the sub-base or the base material, and the cost can be reduced by about 20%. The mechanical properties of stabilized silt was discussed in this paper. Next, the cost-study evaluation will be carried out in combination with the engineering application.
15-The conclusion should be an objective summary of the most important findings in response to the specific research question or hypothesis. A good conclusion states the principal topic, key arguments and counterpoint, and might suggest future research. It is important to understand the methodological robustness of your study design and report your findings accordingly. Please improve your conclusion section.
Response: Thank you for your suggestion. The conclusions has been rewritten to be more concise.
Reviewer 3 Report
This paper presented a study regarding the effect of composite flexible curing agent on aged mechanical properties of treated ilt, in which the replacement levels of the flexible curing agent below 10% by weight are compared. This work introduces new findngs with repsect to the flexible curing agent in cementitious composites. This paper is well written and is in the journal scope. Therefore it is worthy to consider this paper for publication. Hovewer, major revision shoud be introduced into the manuscript. The comments of this paper are as follows:
(1) Keywords. Keywords must indicate the main materials, tests, and methodology used in the study. Therefore, it is required to revise the keywords, add new one, and write based on the points mentioned above.
(2) Abstract. Abstract need to be rewritten to report about the main and new findings obtained in this paper briefly.
(3) Topic selection and literature review. The article concerns the analysis of mechnaical and rheological parameters of composites strenghthening by specific curing agent. This issue should be developed. Furthermore, new curing and nucleating agent materials using in new composites should also be described. Therefore at least, below paper from MDPI database should be discussed and cited in the manuscript:
- "Rheology of cement pastes with siliceous fly ash and the csh nano-admixture”, Materials 2021.
(4) Experiments. Please introduce the name of SEM equipment used in the studies.
(5) Tests. Please provide photos showing specimen during mixing, casting, curing, testing etc. This section should be enlarged and contains relevant photos from the conducted experiments.
(6) Results. Presentation of test results should be shown in different colors. Moreover results on graphs should contain error bars.
(7) Conclusions. Please condense the conclusion part and emphaize the main innovaltive findings.
Author Response
Special thanks to you for your good comments. We have carefully revised the manuscript, and the paper was also re-scrutinized to improve the English. Our responses are as follows, and we amended the relevant part in revised manuscript and the changes below are noted in red words in the article.
1-Keywords. Keywords must indicate the main materials, tests, and methodology used in the study. Therefore, it is required to revise the keywords, add new one, and write based on the points mentioned above.
Response: Thank you for the suggestion. We carefully checked the keywords and confirmed that the keywords in the manuscript can indicate the points mentioned above.
2-Abstract. Abstract need to be rewritten to report about the main and new findings obtained in this paper briefly.
Response: Thank you for the suggestion. we have improved the abstract and made corresponding modification in red words.
3-Topic selection and literature review. The article concerns the analysis of mechnaical and rheological parameters of composites strenghthening by specific curing agent. This issue should be developed. Furthermore, new curing and nucleating agent materials using in new composites should also be described. Therefore at least, below paper from MDPI database should be discussed and cited in the manuscript:
- "Rheology of cement pastes with siliceous fly ash and the csh nano-admixture”, Materials 2021.
Response: Thank you for the suggestion. We have made increase discussion about the analysis of road performance properties of composite curing agent. And the materials using in new composites were described in 2.1 section.
The composite flexible curing agent was developed with sintered red mud and No. 70 base asphalt as the main materials. First, the sintered red mud was dried and ground to less than 120 mesh for use. Then, the base asphalt was heated to a certain temperature and the ground sintered red mud was put into in proportion. After fully stirring for 120s, it was cooled to room temperature. Finally, an appropriate amount of dispersant was put in, and the mixture was crushed to less than 0.075 mm with a pulverizer to obtain the composite flexible curing agent.
The paper concerns the road performance of stabilized silt soil rather than the cement paste. After careful discussion, we think that the paper you mentioned is not suitable for citation in this article.
4-Experiments. Please introduce the name of SEM equipment used in the studies.
Response: The name of SEM equipment used in the study is ZEISS Sigma 500. It has been mentioned in 2.3.5 section.
5-Tests. Please provide photos showing specimen during mixing, casting, curing, testing etc. This section should be enlarged and contains relevant photos from the conducted experiments.
Response: Special Thanks for your suggestion. The relevant photos of experiments are as follows. After careful discussion, we believe that the mixing, casting, curing, and performance testing of the samples are all routine operations according to standards, and there is nothing special, so there is no need to put all the photos in the article.
Specimen casting by static pressure method
The curing of samples
compressive strength test
Test piece and shrinkage meter
Soil sample state after low temperature freeze-thaw
6-Results. Presentation of test results should be shown in different colors. Moreover results on graphs should contain error bars.
Response: Special Thanks for your suggestion. The figures has been modified to show in different colors, and the error bars was also added in related figures.
7-Conclusions. Please condense the conclusion part and emphaize the main innovaltive findings.
Response: Thank you for your suggestion. The conclusions has been rewritten to be more concise.

Reviewer 4 Report
The conducted work “Mechanical Behavior of Yellow River Alluvial Silt Treated with Composite Flexible Curing Agent” is good. However, following comments should be addressed to further improve paper:
- Add more recent relevant literature review from 2021 and 2022 in introduction section. Also, explicitly mention the novelty and research significance of current work in last paragraph of introduction section.
- Avoid long sentences throughout the manuscript, e.g. lines 226-231, etc.
- Show standard deviation where average is being taken.
- Outcome should be further discussed in detail. More scientific reasoning emphasis is required while elaborating outcome.
- There should be a separate section (before conclusions section) explaining the implementation of this research in real field for practicing professionals.
- Closing remarks should be added at the end of conclusion section keeping in mind all conclusive bullet points.
- English Language should be improved throughout the manuscript.
Author Response
I am very grateful to your comments for the manuscript. According with your advice, we amended the relevant part in manuscript. Your questions were answered below, and the changes are noted in red words in the article.
1-Add more recent relevant literature review from 2021 and 2022 in introduction section. Also, explicitly mention the novelty and research significance of current work in last paragraph of introduction section.
Response: Thank you for your suggestion, we have improved the introduction section and made corresponding modification in red words.
Researchers have carried out a lot of experimental research and practical work on the solidification and stabilization materials of silt, and have achieved good theoretical analysis and experimental results. The solidified materials have gradually developed from traditional inorganic and organic types to inorganic-organic composite and biological improvement cured. However, there are still problems of single improvement effect and high technical difficulty in popularization and application of silt solidification in engineering application practice. And research on self-repair is relatively lacking Therefore, it is very necessary to further study the changes of the road performance of the solidified silt under the unfavorable conditions of actual work, such as water, temperature and natural or load effects, on the basis of the basic performance research on the strength and deformation of the solidified silt.
Sintered red mud is a solid industrial waste residue discharged from the production of alumina by the alkaline process. It has high calcium oxide and silicon oxide content, small particles and a network structure inside, which has strong adsorption capacity and certain hydration activity. Asphalt is a complex mixture of hydrocarbons and their derivatives with different molecular weights. It is a temperature-sensitive material with flow self-healing properties. Combined with the characteristics of sintered red mud particles and asphalt materials, the two materials were mixed and ground under a certain process to prepare a new type of silt curing agent, and the silt was solidified and stabilized through physical-chemical comprehensive action. The preparation process of the new curing agent can realize the value-added utilization of industrial solid waste - sintered red mud, and at the same time convert asphalt into solid powder form at normal temperature, which is beneficial to the engineering practice and construction quality control of solidified silt.
On the basis of evaluating the basic mechanical properties of the composite flexible curing agent stabilized silt, the mechanical properties of stabilized silt under test conditions such as water softening, cyclic heating, low temperature freeze-thaw, and high temperature self-healing were further studied, and the effect of the composite flexible curing agent content on water stability of stabilized silt, frost resistance and damage repair performance was analyzed. Scanning electron microscope (SEM) was used to observe the microscopic morphology of silt, cement stabilized silt and the composite flexible curing agent stabilized silt, respectively, to analyze the influence of different materials on soil structure and pore characteristics, and to explore the mechanism of solidification and stability.
2-Avoid long sentences throughout the manuscript, e.g. lines 226-231, etc.
Response: Special thanks for your good comments. We have carefully revised the term in the manuscript and modified long sentences. Corresponding modification was made in red words.
3-Show standard deviation where average is being taken.
Response: Thank you for your suggestion. The error bars has been added in related test results. Corresponding modification was made in red words.
4-Outcome should be further discussed in detail. More scientific reasoning emphasis is required while elaborating outcome.
Response: Thank you for your suggestion. The previous related researches have been added to increase the discussion in Section 3. Corresponding modification was made in red words.
5-There should be a separate section (before conclusions section) explaining the implementation of this research in real field for practicing professionals.
Response: Special thanks for your suggestion. This paper mainly discusses the road performance laboratory test and result analysis of flexible solidified silt. There are many differences between the test indicators and methods of specific project implementation in real field and laboratory tests. We will write another paper to discuss in detail.
6-Closing remarks should be added at the end of conclusion section keeping in mind all conclusive bullet points.
Response: Thank you for your suggestion. The conclusions has been rewritten to be more concise. And the study and future research topics was added at the end of conclusion section
7-English Language should be improved throughout the manuscript.
Response: Special thanks to you for your good suggestion. We have carefully revised the manuscript, and the paper was re-scrutinized to improve the English. Corresponding modification was made in red words.
Round 2
Reviewer 1 Report
All comments are addressed sufficiently.
Reviewer 3 Report
I have no comments.